

# The impact of the Wenchuan earthquake on early puberty: a natural experiment

Qiguo Lian[1,2], Xiayun Zuo[1], Yanyan Mao[1,2], Yan Zhang[1,2], Shan Luo[3], Shucheng Zhang[4], Chaohua Lou[1,5], Xiaowen Tu[1,5] and Weijin Zhou[1,5]

[1] Key Lab. of Reproduction Regulation of NPFPC, SIPPR, IRD, Fudan University, Shanghai, China
[2] School of Public Health, Fudan University, Shanghai, China
[3] West China Second University Hospital, Sichuan University, Chengdu, Sichuan, China
[4] National Research Institute for Family Planning, Beijing, China
[5] Shanghai Institute of Planned Parenthood Research, Shanghai, China

## ABSTRACT

**Background.** The factors influencing pubertal timing have gained much attention due to a secular trend toward earlier pubertal onset in many countries. However, no studies have investigated the association between the Great earthquake and early puberty. We aimed to assess whether the Wenchuan earthquake is associated with early puberty, in both boys and girls.

**Methods.** We used data from two circles of a survey on reproductive health in China to explore the impact of the Wenchuan earthquake on early puberty , and a total of 9,785 adolescents (4,830 boys, 49.36%) aged 12–20 years from 29 schools in eight provinces were recruited. Wenchuan earthquake exposure was defined as those Sichuan students who had not experienced oigarche/menarche before May 12, 2008. Early puberty was identified as a reported onset of oigarche/menarche at 11 years or earlier. We tested the association between the Wenchuan earthquake and early puberty in boys and girls. Then, subgroup analysis stratified by the age at earthquake exposure also was performed.

**Results.** In total, 8,883 adolescents (4,543 boys, 51.14%) with a mean (SD) age of 15.13 (1.81) were included in the final sample. In general, children exposed to the earthquake had three times greater risk of early puberty (boys, RR [95% CI] = 3.18 [2.21–4.57]; girls: RR [95%CI] =3.16 [2.65–3.78]). Subgroup analysis showed that the adjusted RR was 1.90 [1.19–3.03] for boys and 2.22 [1.75–2.80] for girls. Earthquake exposure predicted almost a fourfold (RR [95%CI] = 3.91 [1.31–11.72]) increased risk of early puberty in preschool girls, whereas the increase was about twofold (RR [95%CI] = 2.09 [1.65–2.64]) in schoolgirls. Among boys, only older age at earthquake exposure was linked to early puberty (RR [95%CI] = 1.93 [1.18–3.16]).

**Conclusions.** Wenchuan earthquake exposure increased the risk of early puberty in boys and girls, and preschoolers were more at risk than schoolchildren. The implications are relevant to support policies for those survivors, especially children, to better rebuild after disasters.

Corresponding authors
Xiaowen Tu, tuxiaowen@sippr.org.cn, tuxwcn@163.com
Weijin Zhou, zw0822@sina.com

## BACKGROUND

Pubertal timing is of great interest to both health providers and the public. There's evidence that early puberty is associated with a broad range of short- and long-term adverse health consequences, especially for adolescent girls (*Stroud & Davila, 2016*). The physical and emotional risks of early maturation include not only depression, anxiety, substance abuse and early sexual debut during adolescence (*Mensah et al., 2013*; *Walvoord, 2010*; *Copeland et al., 2010*), but also reproductive cancers and metabolic syndrome in later life (*Day et al., 2015*; *Walvoord, 2010*; *Copeland et al., 2010*). Pubertal timing is a complex trait affected by gene-environment interactions (*Karapanou & Papadimitriou, 2010*). A deeper understanding of the potential contributors to pubertal timing is essential due to a secular trend toward earlier pubertal onset in many countries (*Papadimitriou, 2016*; *Sørensen et al., 2012*).

In addition to chemical exposures and obesity, emotional stress is another culprit that could jump-start puberty earlier (*Ellis & Garber, 2000*; *Rickard, Frankenhuis & Nettle, 2014*; *Sun et al., 2017*). According to psychosocial acceleration theory, an evolutionary-biological perspective on human development, differences in environmental contexts tend to channel children into divergent developmental trajectories in pubertal timing (*Belsky, Steinberg & Draper, 1991*; *Webster et al., 2014*). The stresses caused by harsh environments regulate reproductive strategies more susceptible to enhance reproductive fitness and maximize the probability of leaving descents (*Pesonen et al., 2008*), including early pubertal timing (*Belsky, 2012*; *Webster et al., 2014*). There is considerable evidence that childhood abuse, family conflict, father absence, and parental mental disorder are associated with earlier puberty in girls (*Ellis & Garber, 2000*; *Rickard, Frankenhuis & Nettle, 2014*). A cohort study also linked household socioeconomic disadvantage to earlier pubertal timing in both boys and girls (*Sun et al., 2017*).

War is another harsh environmental stressor that could influence pubertal timing (*Belsky, 2008*). The 1934–1944 Helsinki Birth Cohort Study in Finland revealed that former Helsinki evacuated girls who were sent to live with foster families in Sweden and Denmark during World War II had earlier menarche than the non-separated cohort members who remained at home (*Pesonen et al., 2008*). Additionally, hurricane and earthquake are two of the most common deadly types of natural disasters that could risk the physical and mental health of the child survivors both short-term and long-term (*Marsee, 2008*; *Zheng et al., 2012*). Based on psychosocial acceleration theory, disaster related stressors could affect pubertal timing (*Belsky, Steinberg & Draper, 1991*; *Webster et al., 2014*). However, no study has investigated the association between the earthquake and early puberty.

The 8.0-magnitude Wenchuan earthquake that occurred on May 12, 2008, in Sichuan Province was the biggest and deadliest earthquake since the 1950 Tibet earthquake and 1976 Tangshan earthquake in China. This disaster impacted over 15 million people, rendered at least 4.8 million homeless, and left 69,197 people dead, 374,176 injured and 18,222 listed as missing (*Fan et al., 2017*). The disaster generated a natural experimental setting for measuring the impact of the earthquake for early puberty among the child survivors of the quake. We hypothesized that exposure to the Wenchuan earthquake was associated

with an increased risk of early puberty, with the younger the age at exposure, the higher the risk of early puberty.

## METHODS

### Study design and participants

The data in the current study are drawn from 2013 and 2015 cycles of a multisite survey on the reproductive health among Chinese middle school students aged 12-20 years from 29 schools in eight provinces. The 2013 survey included Sichuan, Hebei, Shaanxi, Shandong, Heilongjiang, and Guangxi, while the 2015 survey sampled Sichuan, Anhui and Inner Mongolia. The data was collected electronically in student computer labs (*Lian et al., 2018*; *Mao et al., 2018*), and included age at menarche/ejaculation, knowledge, attitude and health risk behaviors about reproductive health. We investigated 9,785 adolescents (4,830 boys and 4,955 girls) in two cycles of the survey and dropped 902 adolescents who had experienced oigarche/menarche before the Wenchuan earthquake because their pubertal timing accessed by oigarche/menarche could not be influenced by the earthquake. The final sample for analysis included 8,883 adolescents (4,543 boys and 4,340 girls).

The institutional review board of Shanghai Institute of Planned Parenthood Research reviewed and approved this study (2012–01). Written consents were obtained from both parent and child before the survey.

### Measures

#### Earthquake exposure

For the present study, which was not originally designed to estimate the individual-level exposure to Wenchuan earthquake, we labeled the students who lived in Sichuan as a nature indicator of the Wenchuan earthquake exposure, and their peers in other provinces as a non-exposure group. The Sichuan students were sampled from Wenchuan country, Mianzhu city, Deyang City, Luojiang country, and Zhongjiang country. The distance from the quake's epicenter (kilometers) is 9.60, 94.23, 100.94, 130.76 and 147.76 respectively.

#### Early puberty

We measured pubertal timing by age at first ejaculation (oigarche) for boys or period (menarche) for girls and defined early puberty as a reported onset of oigarche/menarche at 11 years or earlier (*Kaltiala-Heino et al., 2011*). Oigarche and menarche can be used to measure the pubertal maturation for boys and girls respectively in survey studies, although the self-reported data have been criticized (*Dick et al., 2000*). Many studies preferred to use Tanner stages, evaluated by either trained clinicians or children themselves (*Hayatbakhsh et al., 2009*), however, there is evidence that age at oigarche/menarche and Tanner stages produced consistent results (*Kaltiala-Heino et al., 2011*).

#### Confounders

We identified potential confounders by drawing causal acyclic graphs (DAGs) based on prior knowledge in the exposure-outcome association (*Foraita, Spallek & Zeeb, 2014*). In this natural experiment, it appears that there are no essential confounders. Many factors, including body mass index (BMI), socioeconomic status, family structure, and maternal

age of menarche were not considered to be potential confounders because they had no a priori association with Wenchuan earthquake exposure.

### Statistical analysis

Univariate statistics were used to describe the mean age, the frequencies of the individual level and macro level characteristics, and the rate of early puberty. We examined the association of Wenchuan earthquake exposure with early oigarche and early menarche and calculated the strength of associations that measured by rate ratio (RR) with 95% confidential interval (CI) (*Cummings, 2009*). We also computed RRs and 95% CIs for boys and girls respectively. We also performed subgroup analysis on the age group when children were exposed to the Wenchuan earthquake. The age groups at the time of earthquake were recorded as either younger (preschoolers, age <7 years) or older (schoolchildren, age ≥7 years).

We estimated the strength of association of earthquake exposure with RR because this natural experiment can be understood as a retrospective cohort study that earthquake exposure was identified from the recorded information. Statistical analysis used Stata/SE, version 15.1 (StataCorp LLC, College Station, TX, USA).

## RESULTS

A total of 8,883 adolescents from 29 schools in eight provinces were included in the analysis, of which 4,543 were boys (51.14%), and 4,340 were girls (48.86%) (Table 1). The mean age of the final pooled sample was 15.15 years (standard deviation [SD] =1.81). There were eight schools in Sichuan exposed to the Wenchuan earthquake, and 21 schools in other provinces were unexposed. In the final sample, there were 2,658 earthquake survivors and 6,225 adolescents who did not experience the Wenchuan earthquake. Nearly 14% of the participants were preschoolers and the rest, 86%, were schoolchildren when the Wenchuan earthquake struck.

The rate of early puberty was 2.55% for boys and 10.14% for girls (Table 2). As illustrated in Table 3, the rate of early oigarche was 4.9% for male earthquake survivors and 1.54% for boys who did not experience the Wenchuan earthquake. The RR of the association between earthquake exposure and early oigarche was 3.18 [95%CI [2.21–4.57], $p < 0.001$]. Similarly, Table 3 also presents the rate of early menarche (19.50% for exposure and 6.17% for non-exposure) and the RR between earthquake exposure and early menarche (3.16 [2.65–3.78], $p < 0.001$).

We conducted subgroup analysis for males and females respectively, given that the children's age at the time of the earthquake may serve as an effect-measure modifier on the association between earthquake exposure and early puberty (Table 4). The earthquake exposure appeared to double the risk of early puberty (boys:adjusted RR = 1.90 [1.19–3.03], $P < 0.050$; girls: 2.22 [1.75–2.80], $P < 0.050$). As shown in Table 4, older age at the time of the earthquake was a protective factor that reduced the risk of early menarche more than 50% compared with younger age at the time of the Wenchuan earthquake (preschool girls: RR = 3.91 [1.31–11.72], $p = 0.003$; schoolgirls: RR = 2.09 [1.65–2.64], $p < 0.001$). The association was non-significant among adolescent boys who were exposed to the
**Table 1 Sample characteristics by survey cycle, n (%)**

| Characteristics | Survey cycle | | Total (n = 8,883) |
|---|---|---|---|
| | 2013, five years after the earthquake (n = 5,417) | 2015, seven years after the earthquake (n = 3,466) | |
| Age, Mean (SD) | 15.15 (1.78) | 15.11 (1.86) | 15.13 (1.81) |
| Sex | | | |
| Male | 2,849 (52.59) | 1,694 (48.87) | 4,543 (51.14) |
| Female | 2,568 (47.41) | 1,772 (51.13) | 4,340 (48.86) |
| Wenchuan earthquake | | | |
| Exposure | 967 (17.85) | 1,691 (48.79) | 2,658 (29.92) |
| Non-exposure | 4,450 (82.15) | 1,775 (51.21) | 6,225 (70.08) |
| Cities | | | |
| Exposure | 1 | 1 | 1 |
| Non-exposure | 5 | 2 | 7 |
| Schools | | | |
| Exposure | 2 | 6 | 8 |
| Non-exposure | 15 | 6 | 21 |

**Table 2 Sample characteristics by sex, n (%).**

| Characteristics | Boys (n = 4,543) | Girls (n = 4,340) | Total (n = 8,883) |
|---|---|---|---|
| Age, Mean (SD) | 15.16 (1.84) | 15.10 (1.77) | 15.13 (1.81) |
| Wenchuan earthquake | | | |
| Exposure | 1,366 (30.07) | 1,292 (29.77) | 2,658 (29.92) |
| Non-exposure | 3,177 (69.93) | 3,048 (70.23) | 6,225 (70.08) |
| Early puberty | | | |
| Yes | 116 (2.55) | 440 (10.14) | 556 (6.26) |
| No | 4,427 (97.45) | 3,900 (89.86) | 8,327 (93.74) |
| Age at earthquake | | | |
| <7 years | 617 (13.58) | 586 (13.50) | 1,203 (13.54) |
| ≥7 years | 3,926 (86.42) | 3,754 (86.50) | 7,680 (86.46) |

**Table 3 The crude association between the Wenchuan earthquake and early puberty.**

| Early puberty | Wenchuan earthquake, n (%) | | Crude RR (95% CI) | P value |
|---|---|---|---|---|
| | Exposure | Non-exposure | | |
| Boys | | | 3.18 (2.21–4.57) | $p < 0.001$ |
| Yes | 67 (4.90) | 49 (1.54) | | |
| No | 1,299 (95.10) | 3,128 (98.46) | | |
| Girls | | | 3.16 (2.65–3.78) | $p < 0.001$ |
| Yes | 252 (19.50) | 188 (6.17) | | |
| No | 1,040 (80.50) | 2,860 (93.83) | | |

**Table 4** The association between the Wenchuan earthquake and early puberty, by age at the time of the earthquake.

| | Boys | | Girls | |
|---|---|---|---|---|
| | *n* | RR (95% CI, *p* value) | *n* | RR (95% CI, *p* value) |
| Age at earthquake | | | | |
| <7 years | 617 | 1.73 (0.43-6.91, *p* = 0.425) | 586 | 3.91 (1.31-11.72, *p* = 0.003) |
| ≥7 years | 3,926 | 1.93 (1.18-3.16, *p* = 0.007) | 3,754 | 2.09 (1.65-2.64, *p* < 0.001) |
| Crude | 4,543 | 3.18 (2.21-4.57, *p* < 0.001) | 4,340 | 3.16 (2.65-3.78, *p* < 0.001) |
| M-H combined[a] | 4,543 | 1.90 (1.19-3.03, *P* < 0.050) | 4,340 | 2.22 (1.75-2.80, *P* < 0.050) |

**Notes.**
[a] Mantel–Haenszel combined risk ratio.

Wenchuan earthquake before age seven years (RR = 1.73 [0.43–6.91], $p = 0.425$), but significant among the surviving schoolboys (RR = 1.93 [1.18–3.16], $p = 0.007$)

## DISCUSSION

To our best knowledge, this is the first study to investigate the potential impacts of the Great earthquake on pubertal timing. The 2008 Wenchuan earthquake offers a unique opportunity to explore the effects of the earthquake on the quake survivors' later lives. In this natural experiment, exposure to the Wenchuan earthquake doubled the risk ratio of early puberty (1.90 for boys, 2.22 for girls). What's more, there may be effect modification by children's age at earthquake exposure. Earthquake exposure predicted almost a fourfold increased risk ratio of early puberty in preschool girls, whereas the increase was about twofold in schoolgirls. Similarly, earthquake exposure doubled the risk ratio in schoolboys. However, the association of earthquake exposure with early puberty was not statistically significant in preschool boys. The main reason for the unanticipated result was the sample was too small (Unexposed = 50[Cases = 2]).

Factors that influence pubertal maturation have gained much attention recently because of the secular trend to earlier puberty (*Karapanou & Papadimitriou, 2010*). There are many determinants of early puberty, but none of these serve as potential confounders in the present study because the Wenchuan earthquake was apparently not affected by any of these determinants. The age at earthquake exposure might mediate the impact of earthquake exposure on early puberty as an intermediary step in the causal pathway. On the one hand, younger age at earthquake exposure could prolong the time of the earthquake shock action. Children who experienced the earthquake earlier, on the other hand, may suffer more from earthquake-related stress and trauma, which put them at higher risk for earlier puberty. We found that preschool students were more vulnerable to early puberty when they were exposed to the earthquake, which made a new contribution to psychosocial acceleration theory that the first five to seven years of life might be especially influential in shaping reproductive strategy.

In the opinion of evolutionary-mind thinkers, accelerated somatic development was adaptive under dangerous or unpredictable environments because this tactic increased the chances of dispersing genes across generations before dying (*Belsky, 2012*; *Belsky, Steinberg*

& *Draper, 1991*). The adaptive developmental mechanism that enhances reproductive fitness may have negative consequences on individuals in modern society, including early menarche, early sexual debut, and risk-taking (*Belsky, 2012*; *Ellis & Bjorklund, 2012*).

Considerable research on adverse health consequences of disaster events including earthquakes and hurricanes has focused on mental health (*Carroll & Frakt, 2017*; *Fan et al., 2017*; *Shultz & Galea, 2017*). Our findings revealed the profound impacts of the earthquake on early puberty both in boys and girls. A better understanding of these impacts could inform post-disaster interventions, which are crucial to the well-being of survivors, especially children (*Carroll & Frakt, 2017*). Given that children take their cues from adults, parents/guardians/caretakers should act calmly and confidently when disasters occur (*Carroll & Frakt, 2017*). Pediatricians should screen the health conditions of the young survivors of disasters in the years to come. Policies that provide children with supportive environments and caring relationships are critical in rebuilding their lives after disasters.

The underlying biological mechanisms that earthquakes might influence the early puberty are still barely understood. The stressors, including early childhood adversity and earthquake exposure, can affect not only the hypothalamic-pituitary-adrenal (HPA) and hypothalamic-pituitary-gonadal (HPG) axes individually, but the cross-talk between the two axes, possibly mediating HPA-HPG coupling (*Ruttle et al., 2015*; *Shirtcliff et al., 2015*). The mechanism explains how stressors may trigger pubertal development. Animal studies also suggested that social environment could influence the HPG axis at the molecular level (*Maruska & Fernald, 2011*). However, the role of genomic changes regulating pubertal development remains unclear (*Sun et al., 2017*) and is a promising field (*Lomniczi & Ojeda, 2016*). Studies also suggested that major adversity in early childhood could impair the brain's reward circuits processing threat and stress (*Boyce, Sokolowski & Robinson, 2012*; *Hanson, Hariri & Williamson, 2015*), which might influence the regulation of reproductive hormones, resulting in early-onset puberty (*Boyce, Sokolowski & Robinson, 2012*).

## Strengths and limitations

The primary strength of this study is the clear temporal relationship between Wenchuan earthquake exposure and early puberty, which improved our ability to make a valid causal inference. To avoid temporal ambiguity, we excluded individuals who had experienced oigarche/menarche before May 12, 2008. Also, this study took advantage of the natural experiment generated by the Wenchuan earthquake to investigate the long-term effect of the Great earthquake on early puberty.

Our study has several limitations worth noting. First, the earthquake exposure was not measured at an individual level that could have allowed the higher accuracy of exposure estimates. There was not any information on death nor illness caused by the earthquake in their family members in the present study. Second, we failed to investigate the association of severity of exposure with earlier puberty due to the the small sample size in the most damaged areas (237 in Wenchuan and 303 in Mianzhu). Third, the data on oigarche/menarche were collected by self-report and thus may be prone to recall bias. However, there is much evidence to show a high degree of validity in recalled age

at menarche, especially among adolescents (*Koprowski, Coates & Bernstein, 2001*; *Koo & Rohan, 1997*; *Malina et al., 2015*; *Mao et al., 2018*). Fourth, we did not use Kaplan-Meier curve or Cox proportional hazards modeling because of the lack of data on the full spectrum of ages at puberty. It would be much more informative to show the difference in age at puberty curves graphically. Fifth, the study is susceptible to confounding by time trend, which could have been controlled if we could build in a before/after comparison in both exposed and unexposed communities. However, it is impossible to collect data pre-earthquake because of the unpredictable nature of earthquakes (*Fan et al., 2017*). Last, we did not collect other factors that influence puberty, including BMI, maternal BMI, and early maternal menarche, to ensure the comparability between the exposure and non-exposure groups, which could introduce selection bias and hinder the ability to generalize our findings to a more diverse population.

## CONCLUSIONS

Exposure to the Wenchuan earthquake elevated the risk of early puberty in both boys and girls, and preschoolers are more vulnerable to earthquakes compared with schoolchildren. The findings suggest that the health of children, especially younger children, must remain a focus in the recovery from disasters.

## ACKNOWLEDGEMENTS

The authors thank site coordinators and the students involved for their support. The authors also acknowledge the advice from Dr. Qianxi Zhu.

### Funding

This study was supported by National Key Technology R&D Program of China (No. 2012BAI32B02), National Science and Technology Infrastructure Program of China (No. 2013FY110500) and Innovation-oriented Science and Technology Grant from NPFPC Key Laboratory of Reproduction Regulation (No. CX2017-05). The funders had no role in study design, data collection and analysis, decision to publish, or preparation of the manuscript.

### Grant Disclosures

The following grant information was disclosed by the authors:
National Key Technology R&D Program of China: 2012BAI32B02.
National Science and Technology Infrastructure Program of China: 2013FY110500.
Innovation-oriented Science and Technology.
NPFPC Key Laboratory of Reproduction Regulation: CX2017-05.

### Competing Interests

The authors declare there are no competing interests.

## Author Contributions

- Qiguo Lian conceived and designed the experiments, performed the experiments, analyzed the data, contributed reagents/materials/analysis tools, prepared figures and/or tables, authored or reviewed drafts of the paper, approved the final draft.
- Xiayun Zuo, Yanyan Mao, Yan Zhang, Shan Luo and Shucheng Zhang performed the experiments, contributed reagents/materials/analysis tools, authored or reviewed drafts of the paper, approved the final draft.
- Chaohua Lou and Xiaowen Tu conceived and designed the experiments, performed the experiments, authored or reviewed drafts of the paper, approved the final draft.
- Weijin Zhou conceived and designed the experiments, performed the experiments, contributed reagents/materials/analysis tools, authored or reviewed drafts of the paper, approved the final draft.

## Human Ethics

The following information was supplied relating to ethical approvals (i.e., approving body and any reference numbers):

The institutional review board of Shanghai Institute of Planned Parenthood Research reviewed and approved this study (2012-01).

## Data Availability

The raw data are provided in Supplemental Information 1.

## Supplemental Information

Supplemental information for this article can be found online at http://dx.doi.org/10.7717/peerj.5085#supplemental-information.

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
