# Peer review of "The impact of the Wenchuan earthquake on early puberty: a natural experiment"

_PeerJ, doi:10.7717/peerj.5085_

## Round 0.1 · original submission · Major Revisions

· Academic Editor

Major Revisions

The editor recommends that the authors thoroughly proofread the manuscript. For example, contractions (e.g., the use of "wasn't" in line 240 and ""didn't" in line 246) should be avoided in academic writing.

Please address Reviewer 2's comments about the lack of detail on data collection - this is a valid and important concern.Also their comments about confounders should be taken into account.

Reviewer 1 ·

Basic reporting

The Introduction to this paper is succinct and nicely sets the stage for the work presented. But it is surprising that in lines 61 and elsewhere, psychosocial acceleration theory is referred to, but there is no direct reference to it, so the following MUST be cited in several places:
Belsky, J., Steinberg, L., & Draper, P. (1991). Childhood Experience, Interpersonal Development and Reproductive Strategy: An Evolutionary Theory of Socialization. Child Development, 62, 647-670.
When reference is made to the possibility that war is another harsh environment that might affect pubertal timing on line 70, the following reference should also be cited, as this is where that hypothesis is first advanced, building on the preceding citation:
Belsky, J. (2008). War, Trauma and Children’s Development: Observations from a Modern Evolutionary Perspective. International Journal of Behavioral Development, 32, 260-271.
One final comment, especially given the results: It is never made clear that Belsky et al. (1991) hypothesized that it was the first 5-7 years of life that would prove especially influential in shaping reproductive strategy and thus timing of puberty. That developmental viewpoint should be highlighted in the Introduction and discussed in the discussion. I also think that in the Introduction and/or discussion, it should be made clear what the evolutionary foundations of the adversity-puberty hypothesis was: That in the face of risk of early death or compromised development, accelerated somatic development was adaptive because it increased the chances of reproducing before dying.

Experimental design

The design of this epidemiological work is sound, but I was surprised to see the effort to measure first ejaculation in boys, as I have never seen this measured before. THe authors appropriately note that such self-reported data have been criticized, but if they are also referring when they say this to timing of first period/menarche, I would encourage revision of the wording, as there is plenty of good evidence that age of menarche is quite accurately recalled--even decades later. The same is by no means true of first ejaculation.

Validity of the findings

Findings prove both original and interesting.

Additional comments

This paper indisputably extends the literature and makes, with noted revisions, a solid contribution to the literature.

·

Basic reporting

Clear and unambiguous. Only couple of observations regarding comprehension of what the authors try to say.

1. Line 106-107 and Line 163-165 (last one regarding analysis in boys).

Experimental design

Hypothesis is ambiguos. It is talking about prediction of early puberty, but in fact is giving information about RR. Then, should be stated in terms of "increased RR".

Methodology is limited. No information about how the data was collected. It indicated that was a survey. But, no information about the contend of that survey; or about when, how, who took the survey. Was only regarding age of puberty?. Seems to be very unlikely.

Also, there is not analysis of confounder factors. What about information of early puberty in their parents?. What about BMI?. Despite authors said that socioeconomical status would be not a big confounder, I would like to insist about this possibility. Many other confounders can be listed. Then, article will improve including multivariable analysis.

Validity of the findings

Results are novel. Accordingly, information is valuable. But it needs more analysis.

Statistics are fine. Please check again Table 4. Statistics in children < 7 years seems to be wrong.

It is required to include sample size for Table 4.

Discussion is fine. It may be improved including analysis of potential confounders.

Additional comments

It is a nice work. But, seems to be preliminary analysis. Since, crude RR is given, but it should be analysis in terms also of confounder factors; therefore adjusted RR is more precise.

As written, It should be OK if authors decide to published as short communication.

---

## Round 0.2 · accepted · Accept

· Academic Editor

Accept

I can confirm that your manuscript is in good shape, and addressed the reviewer comments.

We are grateful for your contribution to our journal and hope you consider PeerJ again for your future scholarly work.